# Effect of Supplementation with a Specific Probiotic (*Bifidobacterium bifidum* PRL2010) in Pregnancy for the Prevention of Atopic Dermatitis in Children: Preliminary Results of a Randomized Trial

**DOI:** 10.3390/nu17040673

**Published:** 2025-02-13

**Authors:** Caterina Anania, Viviana Matys, Simona Marra, Daniela De Canditiis, Francesca Olivero, Carlo Carraro, Anna Giugliano, Anna Maria Zicari, Maria Grazia Piccioni

**Affiliations:** 1Department of Maternal, Infantile and Urological Sciences, Sapienza University of Rome, 00161 Rome, Italy; viviana.matys@uniroma1.it (V.M.); simona.marra@uniroma1.it (S.M.); carlo.carraro@uniroma1.it (C.C.); anna.giugliano@uniroma1.it (A.G.); annamaria.zicari@uniroma1.it (A.M.Z.); mariagrazia.piccioni@uniroma1.it (M.G.P.); 2Institute for the Application of Calculus (IAC)—CNR, 00185 Rome, Italy; d.decanditiis@iac.cnr.it; 3Independent Researcher, 00100 Rome, Italy; francescaolivero31@gmail.com

**Keywords:** atopic dermatitis, probiotics, microbiota, pregnancy, allergy, infancy, SCORAD

## Abstract

Background: Atopic dermatitis (AD) is a chronic inflammatory skin disease characterized by the appearance of recurrent eczematous lesions and intense itching. The World Allergy Organization (WAO) suggested the administration of probiotics in pregnant women at high risk of allergies in their children. Objectives: Our study aims to evaluate the role of administering the *Bifidobacterium bifidum* strain PRL2010 during pregnancy and breastfeeding in preventing and/or reducing the severity of AD manifestations in children. Methods: It is a monocentric, randomized, double-blind, placebo-controlled trial with probiotic/placebo administration since the 36th week of gestation to mothers with atopy or a family history of atopy; the effects were evaluated over the first 12 months of the children’s lives. Results: No severe adverse effects due to probiotic intake were reported in our cohort. Although proportionally fewer children with AD were in the probiotic group, the statistical analysis showed no significant differences between the probiotic and placebo groups. However, infants who developed the most severe forms of AD in the probiotic group showed a better clinical course during follow-up compared to those in the placebo group. Conclusions: In conclusion, administering the probiotic *Bifidobacterium bifidum* strain PRL2010 during pregnancy and breastfeeding is safe and potentially beneficial; further large-scale studies may confirm its usefulness in improving the clinical manifestation of AD in children with a family history of atopy.

## 1. Introduction

Atopic dermatitis (AD) is a chronic inflammatory skin disease characterized by the appearance of recurrent eczematous lesions and intense itching [1]. Its chronic-recurrent course results in reduced quality of life for patients and causes a major impact on healthcare resources [2].

The prevalence of AD is around 15–20% in children and 1–3% in adults. In recent decades, its incidence appears to have increased by 2–3 times, especially in industrialized countries. In most cases, the onset of AD occurs during the first years of life (80% of cases) and has spontaneous remission during adolescence (60% of cases). AD, asthma, and allergic rhinoconjunctivitis constitute the atopic triad [3]. During the atopic process, an immunological imbalance in favor of CD4+ type 2 (Th2) helper T lymphocytes (T helper cells) and excessive immunoglobulin E (IgE) production occurs: these alterations are responsible for the onset of clinical manifestations.

AD has a multifactorial pathogenesis characterized by an interaction between the patient’s genetic background, skin barrier alterations, immune system abnormalities, and environmental factors [4].

The first element responsible for the pathogenetic mechanism of AD is damage to the skin barrier, which leads to the development of an inflammatory process and subsequent allergic sensitization [5].

Observational studies demonstrate how a family history of atopy can influence the development of AD, showing a higher concordance rate in monozygotic twins than in dizygotic twins [2]. Thus, the magnitude of concordance rates highlights that environmental factors operate in genetically predisposed people [6].

The environmental factors involved include smoking, climate, living in an urban versus rural environment, obesity, pollution, time of weaning, and type of breastfeeding [2].

Scientific evidence also shows that there is a correlation between the presence of atopy in parents and offspring’s susceptibility to it: an atopic parent is more likely than a nonatopic parent to have a child with atopic conditions, including dermatitis [7].

In addition, in patients with AD, intestinal microbial diversity is decreased, and the concentration of microbes such as *Lactobacillus* and *Bifidobacterium* is significantly reduced [8,9]. These microbial changes are present early on in life, supporting the hypothesis that gut dysbiosis can be considered as one of the primary causes of AD.

Gut microbiota are established from the neonatal period and can interact with the host by influencing its metabolism. Moreover, recent studies have identified the presence of bacterial DNA or bacterial products in the meconium, amniotic fluid, and placenta, challenging the idea of fetal sterility and proposing the existence of microbial colonization as early as in utero. Various factors influence the composition of the microbiota from the uterus to postnatal life, such as maternal microbiome, maternal lifestyle (rural/urban) and maternal diet, antibiotic exposure, method of childbirth (natural or cesarean delivery), antibiotic use, and the type of lactation [10]. The state of dysbiosis causes an altered balance between Th1 and Th2.

Probiotics work by favoring the Th1 response, restoring its balance, and providing protection against allergic diseases [11].

In light of these considerations, the consumption of probiotics is useful in providing beneficial bacteria and making up for gut dysbiosis [12].

Based on the currently available studies, it appears that the supplementation of certain types of probiotics leads to effective results in preventing and reducing the severity of AD; however, these conclusions have yet to be validated [13,14].

The World Allergy Organization (WAO) focuses on the study of probiotics and their role in preventing allergic diseases. The results show that probiotic supplementation does not reduce the risk of allergies in general but can provide benefits in the prevention of eczema, suggesting the beneficial use of probiotics in pregnant women at high risk of allergies in their children [15].

Bifidobacteria are the most numerous bacteria in the human microbiota of breastfed infants [16]. They represent the earliest microbial colonizers of the infant’s gut and are acquired from the mother by vertical transmission, which includes both direct mother–child contact at birth and breastfeeding. Specifically, members of the *Bifidobacterium bifidum* species represent the dominant taxa of the intestinal bacterial community and perform different functions, including adhesion to epithelia, the ability to metabolize host glycans, and to activate immunity [17].

*Bifidobacterium bifidum* PRL2010 deserves special attention: indeed, its genome has undergone sequencing and functional analysis, revealing that it represents a model organism for the study of human–microbe coevolution [18]. In fact, it appears to possess a cross-feeding property that supports the growth of other Bifidobacteria: in vitro tests showed that, when introduced in cultures with other strains, it can increase their growth capacity [17].

In addition to the role that *Bifidobacterium bifidum* PRL2010 plays through the aforementioned microbial cross-talk [17], it also exhibits a role in inflammatory response: in particular, studies carried out on Caco-2 cells (colon adenocarcinoma cells capable of expressing differentiating characteristics typical of mature intestinal cells, widely used in in vitro studies), highlighted how it can modulate the expression of genes for interleukins and cytokine transcription regulators. Specifically, *Bifidobacterium bifidum* PRL2010 can increase the transcription of NFKB and, thus, the expression of IL2 and other cytokines, and, at the same time, it can attenuate a proinflammatory response by downregulating certain chemokines and upregulating genes for defensins and tight junctions.

Therefore, these observations suggest that *Bifidobacterium bifidum* PRL2010 exhibits important immunomodulatory activity during intestinal colonization [18].

This study aims to evaluate the effects of administering the probiotic *Bifidobacterium bifidum* strain PRL2010 during pregnancy and while breastfeeding in the context of AD prevention in children. The effects of its administration were evaluated over the first 12 months of life in children born to atopic mothers or mothers with a family history of atopy.

Furthermore, the study also aimed to evaluate whether this type of supplementation can reduce the severity of AD lesions and associated manifestations, such as itching and dry skin.

## 2. Materials and Methods

### 2.1. Participants and Design

The study took the form of a monocentric, randomized, double-blind, placebo-controlled trial conducted at the Department of Maternal, Infantile, and Urological Sciences at Policlinico Umberto I Hospital in Rome. The trial was registered on www.clinicaltrial.gov (Identifier: NCT06809465). Patient recruitment began with the completion of a questionnaire, followed by screening based on the following inclusion criteria:Aged between 18 and 45 years;The presence of allergic manifestations or a positive family history of atopy;An intention to breastfeed.

Exclusion criteria were as follows: the absence of atopy history in the mother or father of the child, maternal age < 18 or >45 years, no clear intention to breastfeed, birth before the 37th week of gestation, and any complication with the newborn such as necrotizing enterocolitis in the infant, immunodeficiencies, congenital diseases, or chronic/metabolic conditions emerging before or during the first few months postpartum.

The selected participants were divided into two arms, with each patient randomly assigned to one of two groups, designated as Group A and Group B, using a random number generator.

The mothers in Group A received a probiotic containing the strain *Bifidobacterium bifidum* PRL2010. Each sachet of preparation used in this study contained 1 billion CFU. The product also included dextrose, rice starch, and soy as additional ingredients. Proper storage required a temperature between +2 °C and +8 °C, with transportation permitted at temperatures below 25 °C for no more than 48 h. Mothers in Group B received a placebo made of maltodextrins. The sachets of the probiotic and placebo were identical in appearance, smell, and taste to ensure blinding. For both groups, the administration method was the same.

Participants in both arms of the trial were instructed to take one sachet daily, starting at 36 weeks of gestation and continuing until delivery. During the first three months of breastfeeding, the mothers assumed the same dosage. The same dosage was administered to their babies starting at the third month of age and continuing up to the sixth month. The powder from the sachets was dissolved in breast milk (or formula, if breast milk was unavailable) and given to the infants using a spoon or syringe.

Participants were asked to report any discomfort or symptoms experienced during the study. Follow-up assessments were conducted via telephone questionnaires at 3, 6, and 12 months postpartum. The questionnaires recorded the presence of AD lesions in the child and other skin conditions such as xerosis, itching, or persistent cradle cap. In the presence of AD, the child underwent skin prick tests for common aeroallergens (dust mites, pollen, cat and dog hair, etc.) and food allergens (milk, egg, dried fruit, fish); skin prick tests are considered positive if the wheal diameter is ≥3 mm compared to the negative control for each allergen tested. An AD assessment was performed using the Scoring Atopic Dermatitis (SCORAD) index, distinguishing mild (SCORAD < 25), moderate (SCORAD 25–50), and severe (SCORAD > 50) forms of AD [19]. The diagnosis and clinical evaluation of eczema and other allergic conditions was carried out by the pediatric allergist.

### 2.2. Ethics

This study was conducted according to the guidelines of the Declaration of Helsinki and approved by the Ethics Committee of Sapienza University of Rome (project number 6205, protocol 0067/2022, approved 28 January 2022).

### 2.3. Informed Consent Statement

Informed written consent was obtained from all subjects involved in this study.

### 2.4. Statistical Analysis

The statistical analysis was divided into two phases. In the first phase, we described the two groups of enrolled patients, analyzing their features and testing for the absence of significant differences between the two groups. This initial phase aimed to ensure that randomization was effective in eliminating other factors that could influence the manifestations of AD symptoms in children from the two groups. In this phase, continuous variables were represented by their mean and standard deviations, while binary variables were reported as their occurrences (percentages). The differences between the two groups were tested using an unpaired Student’s *t*-test for continuous variables and Fisher’s exact test for binary variables, while for categorical variables with more than two categories, we applied the Chi-square test of independence as recommended by De Canditiis D. in (2019) “Statistical Inference Techniques” [20].

In the second phase of the analysis, we investigated the influence of probiotic intake on the manifestation of symptoms of AD. We examined the association between the following binary variables: the presence or absence of AD symptoms (first binary variable) and the treatment type, probiotic or placebo (second binary variable). Identifying an association between these variables would suggest a potential effect of the probiotic on the incidence of dermatitis symptoms. Fisher’s exact test evaluates the null hypothesis of no association between two binary variables, which, in this context, translated to test whether the incidence of dermatitis symptoms was independent of the treatment type (probiotic or placebo). Specifically, we applied a one-sided (left-tailed) Fisher’s exact test to investigate whether dermatitis symptoms occurred less frequently in the probiotic group compared to the placebo group, reflecting a hypothesis of underrepresentation.

For all hypothesis tests, statistical significance was set at *p* < 0.05. Additionally, we calculated the statistical power of the left-sided Fisher’s exact test, which represents the probability of detecting a true association (i.e., rejecting the null hypothesis) under the specified conditions. Data analyses were conducted using MATLAB R2023a, and the power calculations were performed with G*Power software (version 3.1.9.4).

## 3. Results

### 3.1. Enrolled and Randomized Patients

We enrolled 153 patients with a positive family history of allergic diseases and/or AD. Of these, 79 patients were initially recruited and completed the questionnaire but later declined probiotic administration. The most common reasons for this were as follows:Patients who gave birth before 37 weeks or who had a major obstetrical complication;Patients who resided far away from the Policlinico Umberto I Hospital;Patients who had their own distrust and/or their partner distrusted experimental studies;Patients who took too many drugs during pregnancy.

A total of 74 patients were finally recruited and randomized. Among these, 39 were assigned to Group A (*Bifidobacterium bifidum* PRL2010) and 35 to Group B (placebo). All 74 patients completed gestation. Among the 39 patients in Group A, 2 withdrew from the study before they could start probiotic administration; for the same reason, 1 patient in Group B withdrew from the study. Notably, among the 37 patients in Group A and 34 patients in Group B who continued the study, 8 and 11 patients, respectively, did not continue the follow-up at 3 months, having stopped taking the probiotic after the birth of their child. The most common reasons for this were the following:Forgetfulness of taking the probiotic;A high number of medications to take;A lack of perceived benefit from taking it.

At the 6-month follow-up, seven women (four in Group A and three in Group B) decided to withdraw from the study, and at the 12-month follow-up, just one woman in Group B decided not to take part in the follow-up. Moreover, to date, three children (one in Group A and two in Group B) have not yet reached 12 months of age, so they could not be included in the last follow-up. Therefore, we completed the 12-month follow-up with a total of 41 patients, respectively, including 24 in Group A and 17 in Group B.

Figure 1 summarizes the study design; patients’ characteristics are described and analyzed in Table 1.

As shown in Table 1, the two study groups are homogeneous, and no differences in obstetrical outcomes were highlighted, reinforcing the safety of probiotics administration during pregnancy. Moreover, confounding factors that might influence the effectiveness of probiotics, such as the diet and lifestyle of the mothers, have been investigated in the initial questionnaire, and our results showed no significant differences in terms of diet and lifestyle between the two groups, describing two homogeneous groups.

### 3.2. Follow-Up

No serious or mild side effects were reported for this probiotic throughout the study period, strengthening the safety of probiotic administration. Although the study initially included 74 pregnant women divided into two groups, 52 of them completed the 3-month follow-up for their children, 45 completed the 6-month follow-up, and 41 completed the 12-month follow-up. The following sections concern the two groups across the three follow-up periods.

#### 3.2.1. Follow-Up at 3 Months

At the 3-month follow-up, 29 patients were assessed in Group A: 3 children exhibited clear signs of AD, while the remaining 26 showed no such manifestations. In the placebo group, 23 patients were assessed: 4 children displayed clear signs of AD, and the remaining 19 showed none. A summary of these assessments is provided in the contingency in Table 2.

#### 3.2.2. Follow-Up at 6 Months

At the 6-month follow-up, 25 patients in Group A were assessed: 2 children exhibited clear signs of AD, while the remaining 23 showed none. In the placebo group, 20 patients were assessed: 3 children displayed clear signs of AD, while the remaining 17 did not have any symptoms. A summary of these assessments is provided in the contingency in Table 3.

#### 3.2.3. Follow-Up at 12 Months

At the 12-month follow-up, 24 patients were assessed in Group A: 1 child exhibited clear signs of atopic dermatitis, while the remaining 23 showed none. In the placebo group, 17 patients were assessed: 2 children displayed clear signs of atopic dermatitis, while the remaining 15 did not have any symptoms. A summary of these assessments is provided in the contingency in Table 4.

The percentage of patients showing signs of AD in Group A is lower than in Group B at all follow-up times (respectively 10.71% vs. 17.39% at 3 months, *p*-value = 0.3677; 8% vs. 15% at 6 months, *p*-value = 0.3918; 4.17% vs. 11.76% at 12 months, *p*-value = 0.3700), although these differences are not statistically significant. However, a point should be made regarding the calculated power of the statistical test used. In fact, this value is low at all follow-up timings, depending largely on the small sample size. The statistical power of the test is 0.078, 0.069, and 0.065 at 3, 6, and 12 months, meaning that the probability of detecting a true effect is 7.8%, 6.9%, and 6.5%, respectively. This concept largely reinforces the necessity of a larger sample size, possibly in the context of a multi-center study, in order to obtain more accurate data.

To better summarize our study for each follow-up, the percentages of patients with symptoms for each group are shown in Table 5, along with the corresponding *p*-values obtained using the left-tailed Fisher’s exact test.

As a final endpoint, we analyzed the variation in the SCORAD index in patients who developed symptoms during the study. Specifically, at the 3-month follow-up, there were seven patients with symptoms of AD, three in the probiotic group and four in the placebo group. Of these seven patients, two from the probiotic group and two from the placebo group completed the 12-month follow-up. Notably, one of the two patients in the probiotic group who attended the 12-month follow-up no longer exhibited any symptoms. Although the SCORAD index for children with AD symptoms never exceeded 50 (severe AD), a comparison between patients with the most severe forms of AD in both groups at the same follow-up revealed an evident difference. Specifically, the child in the probiotic group demonstrated a much higher percentage of improvement compared to the child in the placebo group. Defining percentage improvement as the relative difference between the SCORAD index at the end (12 months) and at the beginning (3 months) of the follow-up, the child in the probiotic group achieved an improvement of 66.91%, while the one in the placebo group showed an improvement of 34%.

While this result is qualitative within the scope of our study, it supports the hypothesis of the potential role of probiotics in improving the clinical manifestations of AD.

## 4. Discussion

Our study aimed to evaluate the effects of *Bifidobacterium bifidum* PRL2010 administered since gestation in the context of AD prevention in children up to one year of age with a family history of atopy. Scientific evidence converges on the importance of microbiota alterations (especially gut–skin microbiota) [21,22] as a fundamental element in the pathogenesis of AD, which is why modulators of the intestinal bacterial flora, i.e., with probiotics, were chosen as potential elements in its prevention. New knowledge has shown that microbiota development begins in intrauterine life, supporting supplementation with probiotics as early as pregnancy and breastfeeding [15]. Probiotics given to mothers during pregnancy and breastfeeding reach the fetus first and then the newborn. These bacterial strains exert their action after reaching the child’s gastrointestinal tract. Probiotics have immunomodulatory effects on allergic diseases, balancing Th1/Th2 immune response, enhancing Th1, decreasing Th2 response through the secretion of various cytokines, and eliciting beneficial Treg cells to promote immune tolerance [1]. Several studies have analyzed the role of probiotics in the prevention of AD, producing conflicting data [1]. The results are encouraging, even if they cannot be compared easily, given the diversity in the type, dose, and timing of probiotic administration as well as the period of post-treatment follow-up. Due to the effects on intestinal colonization and its immunomodulatory properties, the probiotic selected for our study was *Bifidobacterium bifidum* PRL2010 [18,22]. This strain, already studied by Bellomo et al. in 2024, is currently considered to have an ecological role with the infant’s gut due to genetic adaptation to the human gut microbiota, an interactive role with the host’s immune system, and a role in preserving mucosal integrity [23]. No severe adverse effects were reported in the probiotic group, confirming that its administration during pregnancy and breastfeeding is safe [24]. The analyses obtained on the recruited group of patients did not reveal a statistically significant difference in terms of the presence/absence of the disease between the probiotic and the placebo groups. However, we obtained encouraging results regarding the entity of clinical manifestations of dermatitis in children treated with *Bifidobacterium bifidum* PRL2010 compared to untreated children. The extent and severity of the lesions showed an improvement among the treated patient at the 12-month follow-up; this supports the fact that probiotics may have a role in improving manifestations of AD in children [22,25], especially when administered early (since pregnancy and breastfeeding) [22,26]. The main limitation of this study is the small sample size of subjects enrolled and the conspicuous number of patients that were lost at follow-up. Despite not demonstrating a role for the probiotic *Bifidobacterium bifidum* strain PRL2010 in preventing the development of AD due to the lack of power of the statistical tests, our study supports its possible effect on the severity and clinical course of the disease, showing an improvement in the clinical manifestations of AD in patients who received probiotic administration. Further studies on a larger scale must be performed to provide more robust results to prove this theory.

## 5. Conclusions

AD is a serious global health issue that requires both appropriate and effective treatment and prevention programs. According to our study, the administration of *Bifidobacterium bifidum* PRL2010 during pregnancy and breastfeeding is safe and may have beneficial effects on the severity and duration of AD. However, this is a pilot study, and the anecdotal evidence should be confirmed on a larger scale, with a more substantial sample size, possibly in the context of a multi-center trial.

Although the indications concerning the administration of probiotics to prevent AD are currently approved by the WAO, further and larger studies are needed to confirm the usefulness of probiotics in improving the clinical manifestation of AD in children with a family history of atopy.

## Figures and Tables

**Figure 1 nutrients-17-00673-f001:**
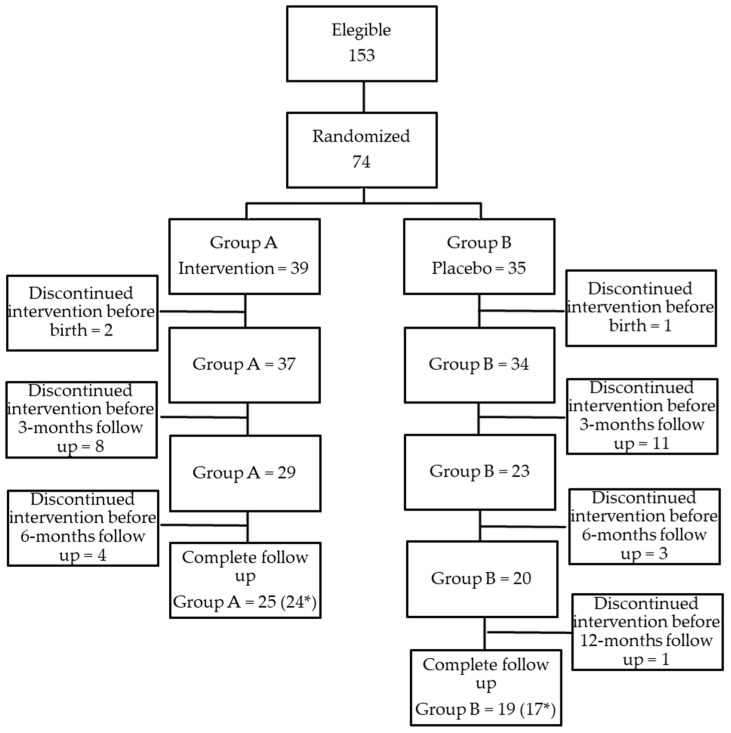
Total patients in our cohort (* number of patients who reached 12 months of age and who completed follow-up at the time of this study’s publication).

**Table 1 nutrients-17-00673-t001:** Study population in the two groups.

Variables	Probiotic Group (A) N = 37 Mean ± SD or n (%)	Placebo Group (B) N = 34 Mean ± SD or n (%)	*p*-Value
Demographic Data			
Age (years)	32.53 ± 5.10	33.56 ± 4.92	0.43283
Smokers	3 (8.1%)	6 (17.6%)	0.2950
Presence of pets	16 (43.2%)	16 (47.1%)	0.8137
History of atopy			
Mother	33 (89.2%)	27 (79.4%)	0.3319
Father	18 (48.6%)	20 (58.8%)	0.4772
Previous child	8 (21.6%)	10 (29.4%)	0.5865
Obstetrical outcomes			
Vaginal birth	16 (43.2%)	17 (50%)	0.63765
Birth weight (g)	3251 ± 498	3334 ± 392	0.4386
Apgar score < 7 at 1’	1 (2.7%)	1 (2.9%)	1
Apgar score < 7 at 5’	0	0	1
Breastfeeding	**N = 28**	**N = 25**	
Exclusive breastfeeding	18 (48.65%)	11 (32.35%)	0.8612
Mixed breastfeeding	6 (16.22%)	8 (23.53%)	
Formula feeding	4 (10.81%)	6 (17.65%)	
Not specified	9 (24.32%)	9 (26.47%)	

**Table 2 nutrients-17-00673-t002:** Comparison between placebo and probiotic groups at 3-month follow-up.

AD Symptoms	Probiotic Group (A)	Placebo Group (B)	Total
No	26	19	45
Yes	3	4	7
Total	29	23	52

Group A: probiotic group; Group B: placebo group; AD: atopic dermatitis.

**Table 3 nutrients-17-00673-t003:** Comparison between placebo and probiotic groups at 6-month follow-up.

AD Symptoms	Probiotic Group (A)	Placebo Group (B)	Total
No	23	17	40
Yes	2	3	5
Total	25	20	45

Group A: probiotic group; Group B: placebo group; AD: atopic dermatitis.

**Table 4 nutrients-17-00673-t004:** Comparison between placebo and probiotic groups at 12-month follow-up.

AD Symptoms	Probiotic Group (A)	Placebo Group (B)	Total
No	23	15	38
Yes	1	2	3
Total	24	17	41

Group A: probiotic group; Group B: placebo group; AD: atopic dermatitis.

**Table 5 nutrients-17-00673-t005:** Percentage of patients with AD in the two groups, *p*-value.

Follow-Up	Group A with AD Symptoms	Group B with AD Symptoms	*p*-Value
3 months	10.71%	17.39%	0.3677
6 months	8.00%	15.00%	0.3918
12 months	4.17%	11.76%	0.3700

Group A: probiotic group; Group B: placebo group; AD: atopic dermatitis.

## Data Availability

The data used in this study are available from the corresponding author upon reasonable request. The data are not publicly available due to privacy reasons.

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
