# Peer review of "Effect of Supplementation with a Specific Probiotic (Bifidobacterium bifidum PRL2010) in Pregnancy for the Prevention of Atopic Dermatitis in Children: Preliminary Results of a Randomized Trial"

_nutrients, 2025, doi:10.3390/nu17040673_

Round 1

Reviewer 1 Report

Comments and Suggestions for Authors

The article is well-written and thorough, presenting promising preliminary results. However, due to the small sample size and participant attrition, further research is needed to confirm the findings. It would also be worthwhile to examine the long-term effects. The study addresses an important topic, as the prevention of atopic dermatitis in children is a highly significant research area. The English text is clear, well-written, and maintains a scientific tone throughout. However, the article does not explain why this particular Bifidobacterium strain was chosen or how it compares to other similar probiotics. Could the use of a different strain or a combination of strains yield more convincing results in the short or long term? It would also be valuable to expand on how maternal lifestyle, diet, physical activity, or other factors might influence the effectiveness of probiotics. Are there any changes in the microbiome composition of the child or mother? How does the probiotic impact the composition of the gut microbiome?

Overall, the article is scientifically well-written, with relevant references, a low plagiarism index, and it addresses an important and timely topic. However, it would be beneficial to expand the discussion, and include an abbreviation key below the tables for clarity.

Author Response

Thank you very much for taking the time to review this manuscript. Please find the detailed responses below and the corresponding revisions highlighted in the re-submitted files:

Minor revision

The article is well-written and thorough, presenting promising preliminary results. However, due to the small sample size and participant attrition, further research is needed to confirm the findings. It would also be worthwhile to examine the long-term effects. The study addresses an important topic, as the prevention of atopic dermatitis in children is a highly significant research area. The English text is clear, well-written, and maintains a scientific tone throughout.

Point 1:

However, the article does not explain why this particular Bifidobacterium strain was chosen or how it compares to other similar probiotics. Could the use of a different strain or a combination of strains yield more convincing results in the short or long term?

Response 1:

The reasons why we investigated this particular strain of Bifidobacterium are synthesized in lines 95-111 of the introduction; in particular, we chose Bifidobacterium because other studies already investigated its prevalence in allergic children (that is reduced), and its properties have already been studied, suggesting that its administration could reduce allergy manifestation and atopic dermatitis. In particular, the strain which was the object of our investigation (Bifidobacterium PRL2010) was already studied in the study of Bellomo et al, who demonstrated its role in preventing atopic manifestations. It was also studied its involvement in colonising infants’ gut and interacting with the host’s immune system, with promising results. We added this information in the discussion (Lines 319-322);

Point 2:

It would also be valuable to expand on how maternal lifestyle, diet, physical activity, or other factors might influence the effectiveness of probiotics. Are there any changes in the microbiome composition of the child or mother? How does the probiotic impact the composition of the gut microbiome?

Response 2:

In our screening questionnaire, we asked the mothers about their lifestyle, physical activity and diet; it showed that no significant differences appeared between the two groups in these terms, reinforcing the fact the groups were homogeneous and these possible confounders did not have any role in influencing the results. We added this data in lines 233-236.

Point 3:

Overall, the article is scientifically well-written, with relevant references, a low plagiarism index, and it addresses an important and timely topic. However, it would be beneficial to expand the discussion, and include an abbreviation key below the tables for clarity.

Response 3:

As you suggested, we expanded the discussion (lines 308-313) and included the abbreviation keys below the tables.

Thank you for your comments and suggestions.

Reviewer 2 Report

Comments and Suggestions for Authors

The randomized, double-blind, placebo-controlled trial aimed to assess the effect of probiotics supplementation (Bifidobacterium bifidum PRL2010) in pregnancy on atopic dermatitis prevention and/or reducing the severity of AD manifestation in children.

Introduction

The introduction contains sufficient information to justify the background of the study. The purpose of the study was clearly formulated.

Material and methods Page 3, line 131 – „The selected participants were divided into two arms, with each patient randomly assigned to one of two groups, designated as Group A and Group B, using a random number generator”. What were the selection (exclusion) criteria (except for information in Page 4, line 159-162)? Please provide more information about the exclusion criteria.

Why was the ingredient used in the placebo (maltodextrin) different from the additional ingredients (dextrose, rice starch, and soy) in the probiotic group?

Results and discussion

Results are clearly presented, and the discussion refers to the results.

As the authors noted, the limitation of the study is the small number of people in groups A and B, which ultimately results in the low power of statistical tests

Author Response

Thank you very much for taking the time to review this manuscript. Please find the detailed responses below and the corresponding revisions highlighted in the re-submitted files:

Introduction

The introduction contains sufficient information to justify the background of the study. The purpose of the study was clearly formulated.

Point 1:

Material and methods Page 3, line 131 – „The selected participants were divided into two arms, with each patient randomly assigned to one of two groups, designated as Group A and Group B, using a random number generator”. What were the selection (exclusion) criteria (except for information in Page 4, line 159-162)? Please provide more information about the exclusion criteria.

Response 1:

The exclusion criteria were the following: absence of atopy history in the mother or father of the child, maternal age < 18 or > 45 years, no clear intention to breastfeed, birth before the 37th week of gestation, any complication of the newborn such as necrotizing enterocolitis in the infant, immunodeficiencies, congenital diseases, or chronic/metabolic conditions emerging before or during the first few months postpartum. We specified the criteria in lines 130-134 for a better understanding.

Point 2:

Why was the ingredient used in the placebo (maltodextrin) different from the additional ingredients (dextrose, rice starch, and soy) in the probiotic group?

Response 2:

We used maltodextrin in the placebo because the company that prepared the placebo for us had this type of substance available.

Thank you for your comments and suggestions.